# Ecological Quality Evolution and Its Driving Factors in Yunnan Karst Rocky Desertification Areas

**DOI:** 10.3390/ijerph192416904

**Published:** 2022-12-16

**Authors:** Shiwen Zhang, Yan Wang, Xuehua Wang, Yang Wu, Chengrong Li, Chao Zhang, Yuhang Yin

**Affiliations:** 1College of Ecology and Environment, Southwest Forestry University, Kunming 650224, China; 2Key Laboratory of Ecological Environment Evolution and Pollution Control in Mountainous Rural Areas of Yunnan Province, Kunming 650224, China; 3College of Forestry, Southwest Forestry University, Kunming 650224, China

**Keywords:** ecological quality evolution, karst rocky desertification, analysis of driving mechanisms, Yunnan karst rocky desertification areas

## Abstract

Rocky desertification is a key element affecting regional ecological quality. Rocky desertification in Southwest China directly affects the ecological security of the Yangtze River and Pearl River basins and also restricts regional economic and social development. In order to clarify the evolution laws and key influencing factors of ecological quality in Yunnan karst rocky desertification areas, a quantitative analysis based on the remote sensing-based ecological index (RSEI) model was conducted to explore the overall evolution characteristics and change laws of ecological quality in Yunnan karst rocky desertification areas in the past 30 years. The correlation between RSEI, rock outcrop rate (Fr), and driving factors was determined by redundancy analysis. The results showed the following: (1) RSEI in Yunnan karst rocky desertification areas generally showed a decreasing trend, with a fluctuation in the mid-term, followed by a tendency to recover. It fell into three stages: decline, trough, and recovery, with fitting coefficients of −0.121, −0.057, and 0.157, respectively. In contrast, Fr showed an opposite tendency, illustrating the inverse relationship between RSEI and Fr, and the rate of sequential succession was much faster than that of the reverse succession under human measures of intervention. (2) The mean value of RSEI of Yunnan karst rocky desertification areas was generally lower than that of the total Yunnan Province land areas and Yunnan non-karst rocky desertification areas, but the mean value of Fr was generally more than that of both the above-mentioned areas. In addition, the RSEI and Fr of Yunnan karst rocky desertification areas both showed lower stability values than those of both the above-mentioned areas. This generally suggested a low ecological quality and a high degree of desertification under a low stability in Yunnan karst rocky desertification areas. (3) The correlation of RSEI and Fr with driving factors followed the order of topographic factors, soil factors > water factors > anthropogenic factors. Anthropogenic factors were the driving force changing the state of rocky desertification, geological factors such as topography and soil to a larger extent determined the original macroscopic ecological relationship of rocky desertification, and water factors lay between the above two. The findings of this research will provide theoretical support and a basis for the improvement of ecological quality and comprehensive control of karst rocky desertification in Yunnan Province.

## 1. Introduction

Due to the karst process of soluble carbonate rocks, the karst landform environment easily leads to the reverse succession of the ecosystem under external interference and forms ecologically fragile areas. Rocky desertification is an extreme manifestation of ecosystem degradation and is the primary ecological problem in karst areas. Globally, soluble carbonates exist on 15.20% of the ice-free surface of continents, with the largest percentage in Europe (21.8%) and the largest absolute area in Asia (8.35 million km^2^) [1]. From the perspective of spatial distribution, the relatively continuous areas of soluble carbonate rocks are mainly distributed in Southwest China–Indochina Peninsula, Europe–Central Asia, and Central America–North America. Meanwhile, there are also vast regions rich in carbonate but discontinuous between Europe and Asia [2]. From the perspective of the population, there are 1.18 billion people living in karst areas of the world, with the highest absolute population in Asia (661.7 million) and the highest percentages in Europe (25.30%) and North America (23.50%) [1]. According to the statistical data from Ford and Williams (2007), about 25% of the global population is partly or entirely supplied by fresh water from karst [3].

The karst rocky desertification areas in Southwest China, with Yunnan–Guizhou–Guangxi as the center, are not only one of the three major regions with a concentrated distribution of carbonate rocks in the world, but also the most fully developed karst continuous belt [4], with the area of more than 540,000 km^2^ and the resident population of 220 million [5]; these areas directly affect the ecological security of the Yangtze River and Pearl River basins and also restrict the regional economic and social development [6]. The third rocky desertification survey in 2016 showed the rocky desertification area of 23,500 km^2^ in Yunnan Province, accounting for 23.40% of the total land area, which is secondary to Guizhou Province in terms of desertification area and has seriously threatened the socioeconomic development and ecological civilization construction of Yunnan Province [7]. In the karst rocky desertification areas, the karst geomorphic environment formed by the development of carbonate rocks provides the material basis for the formation of rocky desertification that is prone to soil and water loss due to the steep and broken karst landform [8,9]. The lack of water–soil environmental elements is the structural defect of the karst ecosystem, and the supply of effective nutrient elements in the soil is insufficient and unbalanced, leading to low ecological capacity and high sensitivity of variation in the karst environment [10]. In addition, karst water is also the dynamic condition from outside for the dissolution of soluble carbonate rocks, which accelerates the development and formation of the karst geomorphic environment under the support of distinct dry and wet climates. Strong interference from human creation is the beginning of a reverse succession of the karst ecosystem and the external cause of human–land contradiction in karst rocky desertification areas, resulting in a vicious cycle of poverty in rocky desertification areas [11,12]. In conclusion, maintaining the stability and balance of geological conditions, soil and water elements, and human activities in karst rocky desertification areas is the key basis of karst rocky desertification control, including further exploration of the influencing mechanisms.

There are studies on the rocky desertification concept [13,14], formation mechanism [15], spatial–temporal evolution [16,17], trends [18], and comprehensive management [19]. Some scholars have also conducted information extraction on rocky desertification [20], explored the interrelationship between rocky desertification and natural and man-made factors [21], simulated the future scenarios of rocky desertification development [22,23], and presented the key technologies and experience of rocky desertification control [24]. Though the above studies are helpful for scientific understanding of the formation, evolution, development, and management of rocky desertification, they only focus on the rocky desertification itself and do not involve the rocky desertification at the level of regional ecological quality which is highly related to human development. Moreover, there is a lack of quantitative evaluation on the impact of the rocky desertification evolution on regional ecological quality. In addition, available studies on ecological quality in Yunnan Province focus mainly on the analysis of vegetation coverage changes [25] and local/urban ecological quality changes [26,27]. There are few analyses on the correlation between rocky desertification and ecological quality in karst ecologically fragile areas in Southwest China through further integration with rocky desertification, the key factor considered as “the source of regional natural disasters, the root of poverty and backwardness, and the hidden risk of ecological security” [28] (pp. 5–6).

In this study, a quantitative approach based on the RSEI model was adopted to analyze the overall evolution characteristics and change rules of ecological quality in Yunnan karst rocky desertification areas during the past 30 years. In addition, the correlation between RSEI, Fr, and driving factors was determined based on the redundancy analysis approach to further explain the impact of rocky desertification on regional ecological quality. This was expected to provide evidence and support for the ecological construction of karst ecological fragile areas in Southwest China.

## 2. Materials and Methods

### 2.1. Study Area

At the boundary between the southern foot of the Qinghai–Tibet Plateau and the Yunnan–Guizhou Plateau, Yunnan Province is located in the southwest of China and is intermingled with high mountains, subalpine mountains, river valleys, and basins. It has a low-latitude plateau monsoon climate, distinct dry and wet seasons, obvious vertical microclimate characteristics, rich biodiversity, and a variety of natural ecological communities from subtropical to alpine cold temperate areas which are the upstream or source of the Yangtze River, Pearl River, Lancang River, and other important international or domestic rivers. The bedrock of the region is mainly limestone, dolomite, and mudstone, in which karstification easily occurs when water is encountered. Being calcium-rich and alkaline, the regional soil mainly includes limestone soil formed by carbonate karst erosion residue and tillage soil formed under long-term cultivation activities, and the supply of effective nutrient elements in the soil is insufficient and unbalanced. The population density of Yunnan province has increased from 93.95 persons/km^2^ (1990) to 126.96 persons/km^2^ (2020), and that of the karst rocky desertification areas increased from 100.83 persons/km^2^ to 139.13 persons/km^2^.

According to the fourth rocky desertification survey by Yunnan Forestry and Grassland Administration in 2021, Yunnan karst rocky desertification areas are found in 88 counties, with a land area of 266,000 km^2^, accounting for more than 67.51% of the provincial land area. They are mainly distributed in the northeast, southeast, and northwest parts of Yunnan, the south part of Northern Yunnan, the east part of Central Yunnan, the southwest part of Southwest Yunnan, and the central part of West Yunnan (Figure 1). These areas mainly feature mountainous landforms, with a widely distributed karst environment. The tectonic movement has brought about steep and broken terrain features. Coupled with the over-carrying of land and the prominent contradiction between man and land, it has resulted in a wide distribution of karst landforms, a serious impact of rocky desertification, and great difficulty in the treatment and control of rocky desertification.

### 2.2. Data Source

The dataset in this study mainly includes time-series Landsat imagery, digital elevation, soil texture, soil depth, regolith and sedimentary deposit thicknesses, soil erosion, annual amount of actual evapotranspiration, annual precipitation, water data, population distribution, spatial distribution of GDP data, road data, and impervious surface data. It also includes some statistical data from *China Desertification Bulletin* and *Yunnan Water Resources Bulletin* (shown in Table 1).

### 2.3. Indicator Extraction

#### 2.3.1. Rocky Desertification Information Extraction

Rock outcrops are important features of a karst rocky desertification area. The rock outcrop rate (Fr), a key index for measuring the degree of rocky desertification, is measured based on dimidiate pixel model to evaluate vegetation coverage in the following equations [20]:(1)$$\mathrm{NDRI}=(\rho_{\mathrm{swir2}}-\rho_{\mathrm{nir}})/(\rho_{\mathrm{swir2}}+\rho_{\mathrm{nir}})$$
where NDRI denotes normalized differential rocky index; ρ_swir2_ and ρ_nir_ denote the infrared 2 band and near-infrared band of Landsat imagery, respectively.
Fr = (NDRI _i_ − NDRI _min_)/(NDRI _max_ − NDRI _min_)(2)
where Fr represents the rock outcrop rate, NDRI _i_ denotes the NDRI value of the i^th^ mixed pixel, NDRI max denotes the maximum NDRI value completely composed of outcrops, and NDRI min denotes the minimum NDRI value completely composed without outcrops. Based on Xu et al. [21], the risk of rocky desertification degree is evaluated and ranked into 6 levels at the 5–95% confidence level: no risk of rocky desertification (level 1, Fr ≤ 20%), latent risk of rocky desertification (level 2, 20% < Fr ≤ 30%), rocky desertification to a light degree (level 3, 30% < Fr ≤ 50%), rocky desertification to a moderate degree (level 4, 50% < Fr ≤ 70%), rocky desertification to a high degree (level 5, 70% < Fr ≤ 90%), and rocky desertification to an extremely high degree (level 6, Fr > 90%).

#### 2.3.2. Ecological Quality Information Extraction

Based on 7 phases of time-series Landsat imagery of Yunnan karst rocky desertification areas from 1990 to 2020, the normalized vegetation index (NDVI), wetness component (WET), index-based built-up index (IBI), bare soil index (SI), and land surface temperature (LST) were obtained by using the bandmath function of ENVI5.X (Figure 2). The outliers beyond the confidence intervals between 0.005% and 99.995% were left out according to the calculation results of the indexes and the actual significance of the data. Due to the difference in the units and orders of magnitude of the above five remote sensing observation indexes, preliminarily synthesized RSEI (RSEI0) was obtained through a fully autonomous principal component transformation after a normalization treatment. Considering the ecological significance of NDVI and WET in the principal component analysis (PCA) results [32], the initial RSEI0 was treated as PC1 in the PCA results. The RSEI imagery of Yunnan karst rocky desertification areas from 1990 to 2020 could be obtained after its normalization. Then, RSEI indexes were graded into 5 levels with an interval of 0.2 based on the research of Xu et al. [33]: excellent (level 5, RSEI > 0.8), good (level 4, 0.8 ≥ RSEI > 0.6), intermediate (level 3, 0.6 ≥ RSEI > 0.4), relatively bad (level 2, 0.4 ≥ RSEI > 0.2), and bad (level 1, RSEI ≤ 0.2). RSEI was used for the further quantification and characterization of the changes in regional ecological quality.

### 2.4. Spatiotemporal Analysis Model

#### 2.4.1. Trend Analysis

The fitting slope was employed to characterize the change rules of RSEI and Fr on the time scale. Its calculation formula is as follows, and it is graded into 5 classes in reference to Wang et al. [34]: a significant decrease (slope ≤ −0.005), a slight decrease (−0.005 < slope ≤ −0.002), basically no change (−0.002 < slope ≤ 0.002), a slight increase (0.002 < slope ≤ 0.005), a significant increase (slope > 0.005).
(3)$$\mathrm{slope}=\frac{n\times \sum_{i=1}^{n}(i\times X_{i})-\sum_{i=1}^{n}i\sum_{i=1}^{n}X_{i}}{n\times \sum_{i=1}^{n}i^{2}-{\left(\sum_{i=1}^{n}i\right)}^{2}}$$
where X*_i_* is the observation index of the *i*th year, *n* is the length of the study time series, and slope is the overall fitting slope of unary linear regression. When the slope is positive, it indicates an upward trend; otherwise, it indicates a downward trend. The larger the absolute value is, the more obvious the change trend is.

#### 2.4.2. Redundancy Analysis

In view of the comprehensiveness and complexity of the ecological environment and its influencing factors, redundancy analysis (RDA) was employed in this study to study the correlation of ecological quality and rocky desertification with key driving factors at the county scale in Yunnan karst rocky desertification areas. The sequence and location of RSEI, Fr, and topography, soil, water, and anthropogenic factors were determined according to their similarity, to explore the interrelationship of ecological quality and rocky desertification degree of sampling plots with driving factors and to quantitatively interpret the effective impacts of a group of variables upon another group [35].

## 3. Results

### 3.1. Evolution Characteristics of Ecological Quality in Yunnan Karst Rocky Desertification Areas

#### 3.1.1. The Overall Evolution Characteristics of Ecological Quality

The calculated RSEI results of Yunnan karst rocky desertification areas from 1990 to 2020 are shown in Figure 3. Combined with the inter-annual variation characteristics, the years 1990 and 2020 witnessed a high ecological quality in the whole area, except for a significant difference observed in the local and regional mean values in Northwest Yunnan only in 1990. The years 2000 and 2010 witnessed a trough in ecological quality. Specifically speaking, the spatial distribution was more reflected in Northeast, Southeast, and South Yunnan and part of Central Yunnan. The years 1995, 2005, and 2015 witnessed a transition characterized by ecological degradation or ecological restoration. The drop in the ecological quality was primarily found in the area ranging from Heqing in West Yunnan to Yibin in Sichuan, from Yongshen to Huping in Northwest Yunnan, in Yuanmou of Central Yunnan, from Zhaoyang district and LuDian county in Zhaotong to Huize and Xuanwei in Qujing City in Northeast Yunnan, from Dongchuan to Xundian in Central Yunnan, from Jianshui to Kaiyuan to Mengzi in South Yunnan, and from Qiubei to Yanshan to Wenshan in Southeast Yunnan. The recovery in ecological quality was primarily found in the area ranging from Diqing Prefecture to Nujing Prefecture to Yulong County, Lijiang City in Northwest Yunnan; from Xishuangbanna to Simao, Lancang, Menglian, and Ximeng in Pu’er City in Southwest Yunnan; and from Jinping, Hekou, and Pingbian in North Yunnan to Maguan, Mali, and part of Xichou in Southeast Yunnan.

#### 3.1.2. Spatiotemporal Variation in Ecological Quality Characteristics

The variation in the mean RSEI value was determined based on seven time series, with 5 years as a time series, to characterize the inter-annual variation in ecological quality characteristics in Yunnan karst rocky desertification areas. A comparison of the variation in the mean RSEI value of karst with the total land areas of Yunnan Province and Yunnan non-karst rocky desertification areas was conducted, as shown in Figure 4. In terms of time scale, the period from 1990 to 2000 witnessed an ecological quality decline. Under continuous disturbance from human activities, RSEI showed a significant downward trend, with a fitting slope equal to −0.121, 24.2 times the threshold of a significant decrease (−0.005). It was 0.002 and 0.008 lower than that in the total Yunnan provincial land areas and Yunnan non-karst rocky desertification areas, respectively. The period from 2000 to 2010 witnessed a trough in ecological quality. Under the interaction between human factors and natural disasters, RSEI fluctuated. The fitting RSEI slopes of the total provincial land areas, karst rocky desertification areas, and non-karst rocky desertification areas of Yunnan were −0.054, −0.057, and −0.046 respectively, all indicating a significant decrease. The period 2010–2020 witnessed a rapid recovery in ecological quality. With the implementation of a series of ecological engineering measures and the incentives of policies, RSEI showed an obvious upward trend, which increased by 0.314 in 10 years, with an average annual growth of 7.22%. The fitting slope was 31.4 times the threshold value of a significant increase (0.005), 0.003 and 0.011 higher than that of the total Yunnan provincial land areas and Yunnan non-karst rocky desertification areas, respectively. From the perspective of the whole time series in this study, the ecological quality of Yunnan karst rocky desertification areas tended to decrease. The overall fitting slope of RSEI variation was equal to −0.009, indicating a significant decrease, 0.001 and 0.004 lower than that of the total Yunnan provincial land areas and Yunnan non-karst rocky desertification areas, respectively.

On the spatial scale, the mean RSEI value in Yunnan karst rocky desertification areas was lower than the provincial mean value, 0.003 (0.38%), 0.002 (0.33%), 0.008 (1.45%), 0.002 (0.30%), 0.014 (3.22%), 0.005 (0.81%), and 0.008 (1.07%) lower in the seven time series, respectively. The largest value difference appeared in 2010, in contrast to the appearance of the smallest value difference gap in 2005 and 1995. The standard deviations were all 0.001–0.004 higher than the provincial standard deviations. This suggested a higher stability for provincial RSEI than for that of karst rocky desertification areas. Similarly, the mean RSEI value of karst rocky desertification areas was 0.009 (1.14%), 0.007 (1.14%), 0.024 (4.36%), 0.006 (0.91%), 0.047 (10.80%), 0.016 (2.60%), and 0.025 (3.34%) lower than those of non-karst rocky desertification areas in the seven time series, respectively. The largest value difference emerged in 2010, in contrast to the appearance of the smallest value difference in 2005. The standard deviations were all 0.002–0.021 higher than those of the non-karst rocky desertification areas, indicating a higher RSEI stability in the non-karst rocky desertification areas than in the karst rocky desertification areas.

#### 3.1.3. Hierarchical Change Characteristics of Ecological Quality

According to the classification standard of the ecological level every 0.2, the RSEI of Yunnan Karst Rocky Desertification Areas during the past 30 years was graded from the pixel scale (Table 2), so as to further discuss the graded change in the ecological quality of Yunnan karst rocky desertification areas.

From the perspective of ecological grade, the rapid decline in ecological quality of Yunnan karst rocky desertification areas from 1990 to 2000 mainly showed the continuous decline in ecological quality at a high level, from level 4–5 (95.40%) in 1990 to level 3–4 (99.97%) in 1995 and level 3–4 (95.72%) in 2000. In these areas, 66.23% of level 4 (1995) was more than others, while 64.31% of level 3 (2000) was more. The area of ecological condition deterioration grade decline was 253,900 km^2^, accounting for 95.23%, of which the proportion of level -1 and -2 decline was 25.30% and 62.83%, respectively.

From 2000 to 2010, the ecological quality of Yunnan karst rocky desertification areas was the lowest stage, which fluctuated between the two historical lows of the regional mean of RSEI, mainly recovering from level 3–4 in 2000 (95.72%) to level 3–4 in 2005 (99.69%). Under the influence of three consecutive years of drought (2009–2011), it fell again to level 2–3 in 2010 (90.26%). In these areas, 64.31% of level 3 (2000) was more than others, while 80.03% of level 4 (2005) was more and 51.66% of level 3 (2010) was more. The area of declining ecological condition deterioration grade was 246,000 km^2^, of which level -1 (79.21%) accounted for the majority, while the area of rising ecological condition improvement grade was only 20,200 km^2^, accounting for 7.58%.

After 2010, the ecological quality of Yunnan karst rocky desertification areas was in the continuous recovery stage, mainly from level 2–3 in 2010 (90.26%) to level 3–4 in 2015 (97.11%) and to level 4–5 in 2020 (87.11%). The area with improved ecological conditions was 260,700 km^2^, accounting for 97.79% of the total, of which 13.15%, 60.87%, and 23.32% were in level 1–3, respectively. Compared with the ecological quality in the early 1990s, Yunnan karst rocky desertification areas still had some room for improvement, while the proportion of the area with improved ecological conditions was only 39.27%.

### 3.2. Evolution Characteristics of Rocky Desertification Degree in Yunnan Karst Rocky Desertification Areas

#### 3.2.1. The Overall Evolution Characteristics of Rocky Desertification Degree

Fr is a key indicator for measuring rocky desertification degree. That is to say, different Fr values mean different rocky desertification degrees, namely different land degradation degrees. The calculated results of Fr in Yunnan karst rocky desertification areas from 1990 to 2020 are shown in Figure 5. Regarding the spatial distribution characteristics, more prominent Fr values were observed in Northeast, Southeast, and South Yunnan and part of Central Yunnan among the karst rocky desertification areas in the process of comprehensive management of rocky desertification. Specifically speaking, more attention should be paid to the regions along the line from Zhaoyang–Ludian–Huize in Northeast Yunnan to Dongchuan–Xundian in Central Yunnan, from Jianshui–Kaiyuan–Mengzi in South Yunnan to Qiubei–Yanshan–Wenshan in Southeast Yunnan. By contrast, the vegetation showed a rather good coverage in the regions along the line from Diqing Prefecture–Nujiang Prefecture to Yulong County in Lijiang City in Northwest Yunnan, from Xishuangbanna to Simao–Lancang–Menglian–Ximeng in Pu’er City in Southwest Yunnan, from Jinping–Hekou–Pingbian in South Yunnan to Maguan–Mali–Xichou in Southeast Yunnan, from Fuyuan–Luoping–Shizong in Northeast Yunnan to Guannan–Funing in Southeast Yunnan. Accordingly, Fr in these regions features a rather low level of desertification degree among karst desertification areas.

#### 3.2.2. Spatiotemporal Variation in Rocky Desertification Degree

The variation in the mean Fr value was determined based on imagery from seven time series, with 5 years as a time series, to characterize the inter-annual variation in rocky desertification degree in Yunnan karst rocky desertification areas. A comparison of the variation in the mean Fr value between karst and non-karst rocky desertification areas of Yunnan Province was conducted, as shown in Figure 6. The mean Fr values of Yunnan karst rocky desertification areas from 1990 to 2020 were 0.427, 0.483, 0.541, 0.477, 0.537, 0.523, and 0.517, generally taking on an M-shaped trend until they tended to be stable. They peaked twice in 2000 and 2010 respectively. The overall regional mean value was about 0.5, suggesting a rocky desertification degree between mild and moderate, which was consistent with the conclusion of the overall rocky desertification degree in Yunnan Province in the third rocky desertification survey in 2016 [36] (pp. 25–42).

On the time scale, the period from 1990 to 2000 witnessed worsening rocky desertification in Yunnan karst rocky desertification areas. Under continuous disturbance from human activities, Fr showed obvious growth. The fitting slope was equal to 0.057, 11.4 times the threshold value of a significant increase (0.005), 0.019 and 0.063 higher than that of Yunnan province average and non-karst rocky desertification areas, respectively. From 2000 to 2010, the degree of rock desertification fluctuated. Under the interactive influence of human factors and natural disasters, Fr showed a fluctuating tendency. Fr values of the total Yunnan provincial land areas and Yunnan non-karst rock desertification areas both suggested a significant decrease, while Fr of Yunnan karst rock desertification areas suggested a slight decrease. The period from 2010 to 2020 witnessed a continuous decrease in rock desertification degree. Under the implementation of a series of ecological engineering measures and incentive policies, Fr showed a gradual downward trend, decreasing by 0.02 in 10 years, with an average annual decrease of 10.0%. The fitting slope was 2.0 times the threshold value of a significant decrease (−0.005), 0.011 and 0.035 lower than that of the total Yunnan provincial land areas and non-karst rocky desertification areas.

On the spatial scale, the mean Fr value in Yunnan karst rocky desertification areas was higher than that of the provincial mean value, −0.005 (−1.17%), 0.012 (2.48%), 0.034 (6.28%), 0.016 (3.35%), 0.048 (8.94%), 0.017 (3.25%), and 0.027 (5.22%) higher in the seven time series, respectively. The largest value difference appeared in 2010, in contrast to the appearance of the smallest value difference in 1990. The standard deviations were all −0.001, 0.001–0.005 higher than those of the provincial standard deviations. This suggested a higher Fr stability in the total provincial land areas than in karst rocky desertification areas. Similarly, the mean value of Fr in karst rocky desertification areas was −0.017 (−3.98%), 0.042 (8.70%), 0.109 (20.15%), 0.052 (10.90%), 0.157 (29.24%), 0.057 (10.90%), and 0.086 (16.63%) higher than that in non-karst rocky desertification areas in the seven time series, respectively. The largest value difference emerged in 2010, in contrast to the appearance of the smallest value difference in 2005. The standard deviations were all −0.004, 0.007–0.038 higher than those of the non-karst rocky desertification areas, indicating a higher stability of Fr in the non-karst rocky desertification areas than in the karst rocky desertification areas.

#### 3.2.3. Hierarchical Change Characteristics of Rocky Desertification Degree

According to the relevant standards, the classification of rocky desertification in Yunnan karst rocky desertification areas was completed according to Fr, for which the classification statistic was carried out from the pixel scale (Table 3), so as to further discuss the classification change of the degree of rocky desertification in Yunnan karst rocky desertification areas.

The period from 1990 to 2000 was the stage of the intensification of the degree of rocky desertification in Yunnan karst rocky desertification areas, which was mainly manifested as the continuous increase in the area of level 4–6 rocky desertification, which was from 34.32% (1990) to 46.31% (1995) and then to 56.04% (2000). The total area was increased by 57,900 km^2^, of which level 6 increased by 6.50% with 17,300 km^2^.

From 2000 to 2010, the degree of rocky desertification in Yunnan karst rocky desertification areas fluctuated between the two extremes of the mean value, which was mainly manifested as levels 4–6 decreased from 56.04% (2000) to 46.04% (2005). Under the influence of three consecutive years of drought (2009–2011), it recovered to 54.88% in 2010. The total area decreased by 3100 km^2^, among which the area of level 6 increased by 1200 km^2^.

Since 2010, the degree of rocky desertification in Yunnan karst rocky desertification areas has been alleviated continuously, mainly showing that levels 1–3 increased from 45.13% (2010) to 47.15% (2015) and then to 50.28% (2020), with the increase in total area by 13,700 km^2^. Meanwhile, levels 4–6 continued to decline to varying degrees, decreasing by 1300 km^2^, 4800 km^2^, and 7700 km^2^, respectively. Compared with the grade distribution of rocky desertification in the early 1990s, the relevant control measures and policies of Yunnan karst rocky desertification areas should be promoted continuously.

### 3.3. Correlation between Ecological Quality Evolution and Driving Factors

#### 3.3.1. Correlation Analysis with Modeling Factors

The correlation between RSEI, Fr, and modeling factors from 1990 to 2020 is shown in Figure 7. The first and second ranking axes can explain more than 90% of the relationship of modeling factors with RSEI and Fr. Five time series, namely 1990, 1995, 2010, 2015, and 2020, accounted for more than 95% of the variation. The contribution degree of modeling factors was in the order of NDVI > WET > IBI, LST > SI. Most indexes contributed at the significance level of *p* < 0.01, except SI (1995, 2005) and LST (2020) did not pass the significance test at the level of *p* < 0.05. In 1990, RSEI and Fr were close to orthogonal, but from 1995 to 2020, they showed a relatively stable obtuse angle, indicating little correlation between RSEI and Fr and almost no impact of rock desertification on regional ecological quality in the early 1990s. As they showed a stable negative correlation, rocky desertification had become a key negative index affecting regional ecological quality. The NDVI and WET were positively correlated with RSEI, with NVDI showing a larger correlation. The total of NDVI and WET contributed more than 70% (*p* < 0.01, highly significant); in particular, in 2005 and 2020, they contributed more than 90%. The IBI and SI were positively correlated with Fr, and their correlation fluctuated within a certain range. IBI showed an extremely significant correlation with Fr (*p* < 0.01), while SI showed an unstable correlation with Fr, namely sometimes significant, sometimes extremely significant, and sometimes insignificant. LST has a certain impact on RSEI and Fr. LST was positively correlated with RSEI but negatively correlated with Fr. It contributed less than 20% at the significance level of *p* < 0.01 (extremely significant), suggesting an obvious limited impact.

#### 3.3.2. Correlation between Ecological Quality and Topographic and Soil Factors

The correlation of RSEI and Fr with topographic and soil factors from 1990 to 2020 is shown in Figure 8. The first and second ranking axes can explain more than 20–40% of the correlation of topographic and soil factors with RSEI and Fr. Among the topographic factors, elevation, slope gradients, and slope aspect were positively correlated with RSEI (except for the year 1990) but negatively correlated with Fr (except for the year 2000). No obvious rules were observed on the correlation between elevation and Fr. The total contribution of topographic factors ranged between 20 and 35%. In terms of soil texture factors, Fr was positively correlated with pH value, soil bulk density, and soil organic carbon, with the correlation order of pH > soil bulk density > soil organic carbon, but it was negatively correlated with soil available water content. RSEI was negatively correlated with pH value and soil bulk density (except for the year 1990). The correlation between RSEI and soil organic carbon shifted gradually from a negative correlation to a positive correlation after an orthogonal intersection. A rather large but irregular fluctuation was observed in the correlation between RSEI and soil available water content. Generally speaking, the total contribution of geological factors ranged from about 15% to 60%. The minimum contribution appeared in 1990, scoring 14.8%, and the maximum contribution appeared in 2000, scoring 60.9%. Among the other soil factors, regolith (except for the year 2020), soil erosion, and soil layer (except for the years 1995 and 2005) were positively correlated with Fr, and a shift from a negative correlation to a positive correlation was observed between sedimentary deposit layer and Fr. The regolith (except for the years 2010 and 2020) and soil erosion (except for the year 1990) were negatively correlated with RSEI; the soil layer tended to shift from a positive correlation with RSEI to a negative correlation, in contrast to a shift from a positive correlation to a negative correlation observed between sedimentary layer and RSEI. The total contribution of other soil factors ranged from 15 to 55%, with a rather large variation span, among which regolith and soil erosion contributed the most.

#### 3.3.3. Correlation of Ecological Quality with Water and Anthropogenic Factors

The correlation of RSEI and Fr with water factors from 1990 to 2015 is shown in Figure 9. The first and second ranking axes can explain less than 20% of the correlation of water factors with RSEI and Fr. The contributions of individual water factors varied greatly, ranging from 78.9% to 2.8%. Water surface density contributed the most in 2005, while actual evapotranspiration contributed the least in 1995. Fr was negatively correlated with annual precipitation, water body area, and water surface density, while the correlation with annual actual evapotranspiration varied greatly and showed no obvious regularity. RSEI was positively correlated with annual precipitation (except for the year 1995), water body area, and water surface density (except for the year 1990), and the correlation was roughly in the order of annual precipitation > water body area > water surface density; RSEI was negatively correlated with annual actual evapotranspiration (except for the year 1990).

The correlation of RSEI and Fr with anthropogenic factors from 1990 to 2015 is shown in Figure 10. The first and second ranking axes can explain less than 15% of the correlation of anthropogenic factors with RSEI and Fr. Among anthropogenic factors, population spatial distribution generally contributed greatly; in contrast, other anthropogenic factors tended to make an unstable contribution. The contribution of GDP spatial distribution gradually exceeded that of population spatial distribution. The total contribution of road length and road surface density stayed at about 20%. Fr is positively correlated with such anthropogenic factors as population spatial distribution (except for the years 1995 and 2005), GDP spatial distribution (except for the year 2005), road length (except for the years 2000 and 2005), road surface density (except for the year 2005), impervious surface area (except for the years 1995 and 2005), and impervious surface density (except for the year 2005). Except for the years 1995 and 2005, RSEI was negatively correlated with the main anthropogenic factors but positively correlated with road length (except for the year 2015). Accordingly, the impacts of anthropogenic activities on the karst rocky desertification areas in Yunnan Province tended to be partly positive and mostly negative.

## 4. Discussion

### 4.1. Ecological Quality Evolution in Karst Rocky Desertification Areas and Its Causes

Based on the remaining native vegetation communities and relevant historical literature, the conclusion can be reached that the karst areas of Yunnan Province were originally dominated by a vegetation landscape composed of wet broadleaved forest, semi-wet broadleaved forest, and limestone mountain shrub, while the current rocky desertification landscape was the result of a gradual evolution through degradation under the continuous and strong influence of human activities [37] (p. 85). Specifically, the increase in rural population and the enhanced dependence on land, coupled with unreasonable reclamation methods, deforestation, and land reclamation, had led to a massive reduction in land vegetation cover, aggravated soil erosion, and large areas of outcrops [16]. According to the calculation scenarios of RSEI under different ecological conditions [33], under the scenario that it is higher than 0.63 in areas with dense forests or vegetation or as low as 0.18 in areas with severe soil erosion, when soil erosion exceeded the maximum allowable values, rocky desertification tended to become irreversible [38]. This may indicate the important role of land cover in maintaining ecological quality, and its greater role in karst ecosystems with low ecological carrying capacity, high sensitivity to variation, and weak disaster-bearing capacity [39] (pp. 16–22). This is consistent with Cao et al.’s [40] conclusion that the fragile karst environment, combined with population pressure and unreasonable production activities, caused worsening rocky desertification in Southwest China from 1987 to 2005.

In 1999, the national project “returning farmland to forest or grassland” was initiated. In 2000, the control and management of rocky desertification were included in “the 10th-Five-Year Plan” [18]. Since 2006, a demonstration exploration of typical rocky desertification control has been carried out [41]. Later followed the reform on the forest rights of collective-owned forests and the rocky desertification comprehensive treatment. Since the 18th National Congress of the Communist Party of China, a series of ecological restoration policies and ideas have been proposed and implemented, such as the advancement of ecological progress and “building beautiful China”. In addition, the urbanization process has caused a large proportion of the rural population to migrate into cities as a workforce. Consequently, much cultivated land has been abandoned, and the conflict between humans and land has been alleviated to a certain degree. In addition, the reform of traditional energy mode has also reduced people’s dependence on firewood, and the vegetation cover has been gradually restored, resulting in a gradual alleviation of the rocky desertification degree [36] (pp. 78–84). In 2010, ecological deterioration occurred, involving a decline in ecological quality and the deepening of rocky desertification to different degrees in all areas of Yunnan province. Due to the fragility of karst environments, in the karst rocky desertification areas in Yunnan province, the ecological deterioration is more prominent. Kunming City is a typical karst area in Central Yunnan and even in the whole region, and its ecological quality in this study is similar to the findings by Zhao et al. [42], who drew the conclusion that the ecological quality was “extremely unsafe” through the analysis on the relationship between the ecological environment and the development of Kunming City from 2000 to 2010 based on an ecological footprint model. It is also consistent with Nong et al.’s [26] ecological quality dynamical monitoring based on the RSEI model in Kunming.

A decrease in RSEI by 0.227 and an increase in Fr by 0.06 were observed in karst rocky desertification areas in Yunnan Province in 2010 compared with 2005. This broke the trend of continuous improvement in ecological quality since 2000. In these areas, the area with ecological deterioration decreased by 261,500 km^2^. The newly increased level -1 and level -2 rocky desertification areas accounted for 34.38% and 61.37%, respectively; the newly increased level 4–6 rocky desertification areas amounted to 23,600 km^2^, accounting for 8.84% of the total provincial land area, and the level 6 rocky desertification areas increased by 11,900 km^2^. The main reason lies in the continuous expansion of rocky desertification areas in the Pearl River Source in Qujing, Yunnan Province, as a result of the consecutive three-year drought (2009–2011) [43]. It is also embodied in the statistical data of precipitation in *Yunnan Water Resources Bulletin (2009/2010/2011)*. The evolution trend of the rocky desertification degree in the karst rocky desertification areas of Yunnan Province from 2005 to 2010 in this study is contrary to the conclusion of Luo et al. [16] who considered that the karst areas in Southwest China during the same period tended to decrease. The reason might lie in the continuous drought which had made the Pearl River Source in Qujing, Yunnan Province, one of the key monitoring areas in the second rocky desertification survey, the only one with the expansion of rocky desertification areas [43].

### 4.2. Correlation between Ecological Quality and Driving Factors in Karst Rocky Desertification Areas

The correlation of NDVI, WET, IBI, SI, and LST with regional ecological quality was verified by RDA. Among them, NDVI and WET tended to be positive and contributed a large part, which indicated the mechanism for the key role of vegetation restoration in ecological restoration and reconstruction in the rocky desertification areas of Southwest China [44]; IBI, SI, and LST tended to be negative, which was consistent with the research conclusion of Li et al. [32], who used eigenvectors in PCA to determine the tendency of observation indicators. In addition, the correlation between RSEI and Fr suggested a stable negative correlation between the regional ecological quality and rock desertification, which was consistent with observations and general experience.

In the correlation of RSEI and Fr with topographic, soil, water, and anthropogenic factors in the six to seven time series, the correlation order of topographic factor, soil factor > water factor > anthropogenic factor was determined based on their interpretation of variations in the first and second axes. It further verified the primary role of geological factors in determining the macro-ecological pattern of rocky desertification in Southwest China [9]. Xu et al. [21] analyzed the driving factors of rocky desertification in typical karst areas of Yunnan Province and concluded that population density, GDP, and annual precipitation were the main driving factors for the formation of rocky desertification, with population density contributing the most and the combination of GDP and population density playing a rather large role in the multi-factor interaction. Xu et al.’s conclusion was further supported by the correlation with water and anthropogenic factors in this study. Chen et al. [45] explored the expansion mechanism of rocky desertification based on stalagmite $\delta^{13}C$ in Shijiangjun Cave, Guizhou Province, and argued that the joint action of human activities and climate change led to the expansion of rocky desertification; Du et al. [19] concluded that rocky desertification in Southwest China is the process or result of ecosystem degradation under the action of human–land conflict. However, anthropogenic factors are the driving force changing the state of rocky desertification [6], and geological factors are the ones determining the original macro-ecological relationship of rocky desertification [9]; namely, the fragile ecological and geological environment is the basis for the formation of rocky desertification, and the role of water factors lies between those of geological factors and anthropogenic factors. This is in conflict with traditional public awareness to a certain degree, which requires more detailed research and exploration.

## 5. Conclusions

The overall ecological quality of Yunnan karst rocky desertification areas tended to be W-shaped and fell into three stages, namely decline, trough, and recovery, which was in contrast to the evolution of the rocky desertification degree. The correlation analysis showed that the rocky desertification degree of karst rocky desertification areas negatively affected the stability of the regional ecological quality, and a significant decrease in the correlation between rocky desertification degree and the stability of the regional ecological quality was observed in non-karst rocky desertification areas. A lower ecological quality or a higher degree of rocky desertification under low stability was observed in Yunnan karst rocky desertification areas as a whole than in the total provincial land areas and non-karst rocky desertification areas. This may indicate a worse ecological quality, a deeper rocky desertification degree, a higher variance-based sensitivity, and a larger variation range in Yunnan karst rocky desertification areas. The correlation of RSEI and Fr with driving factors followed the order of topographic factors, soil factors > water factors > anthropogenic factors. Anthropogenic factors were the driving force changing the state of rocky desertification, and geological factors, such as topography and soil, tended to determine the original macro-ecological relationship of rocky desertification, with water factors lying between them.

The study focuses on the evolution trend, spatial distribution characteristics, and key driving factors of the ecological quality and the rocky desertification degree in Yunnan karst rocky desertification areas, which were compared with the total provincial land areas and non-karst rocky desertification areas. In a follow-up study, the time scale and spatial scope can be further refined, with a focus on typical regions and typical river basins, and the driving mechanism can be deeply analyzed to further expand the depth and breadth of the research results. From a global perspective, our findings will have an outstanding guiding significance for the control of rocky desertification in the Indochina Peninsula preferentially. After all, Indochina Peninsula is directly adjacent to China, and the karst development degree has continuity in geographical space. Karst rock desertification is not extremely prominent in soluble carbonate rock salt distribution areas of Europe and Central America, but the finding is of great significance for understanding the vulnerability of the karst ecosystem and the development mechanism and environmental effects of rock desertification, so as to reduce the driving force of human activities on the retrogressive succession of karst ecosystems.

## Figures and Tables

**Figure 1 ijerph-19-16904-f001:**
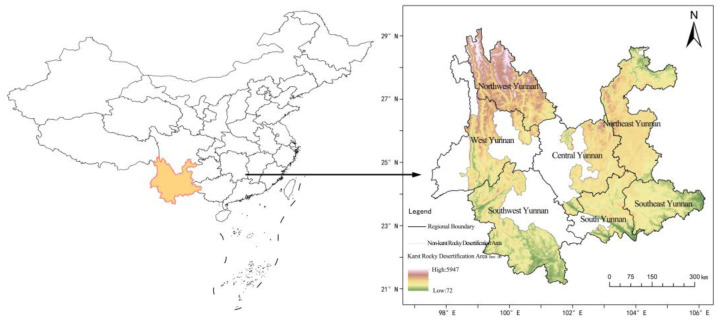
Location of the study area and the area division.

**Figure 2 ijerph-19-16904-f002:**
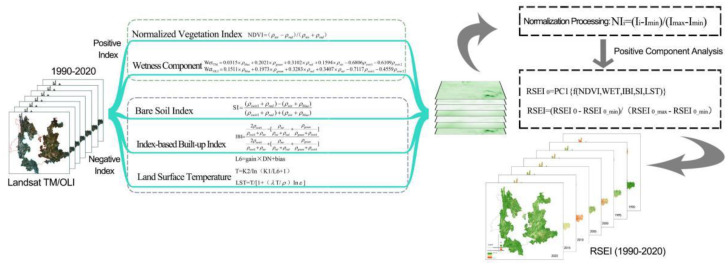
Steps for the extraction of ecological quality information (RSEI).

**Figure 3 ijerph-19-16904-f003:**
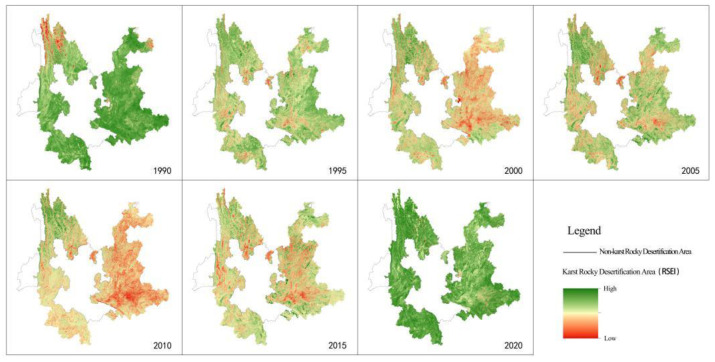
Dynamic evolution characteristics of RSEI in Yunnan karst rocky desertification areas (1990–2020).

**Figure 4 ijerph-19-16904-f004:**
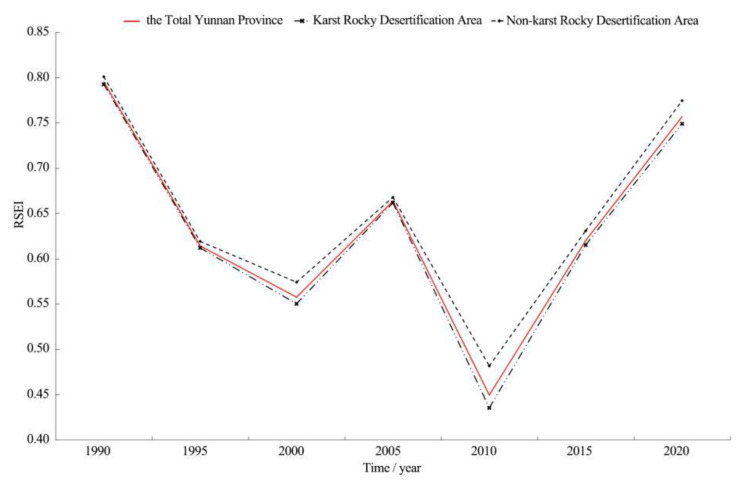
Statistics of changes in RSEI mean value in Yunnan Province (1990–2020).

**Figure 5 ijerph-19-16904-f005:**
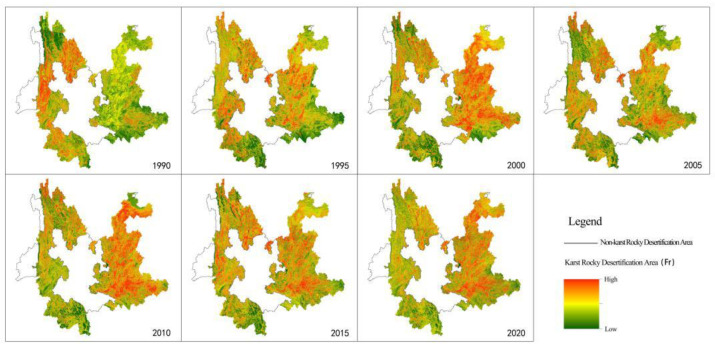
Dynamic evolution characteristics of Fr in Yunnan karst rocky desertification areas (1990–2020).

**Figure 6 ijerph-19-16904-f006:**
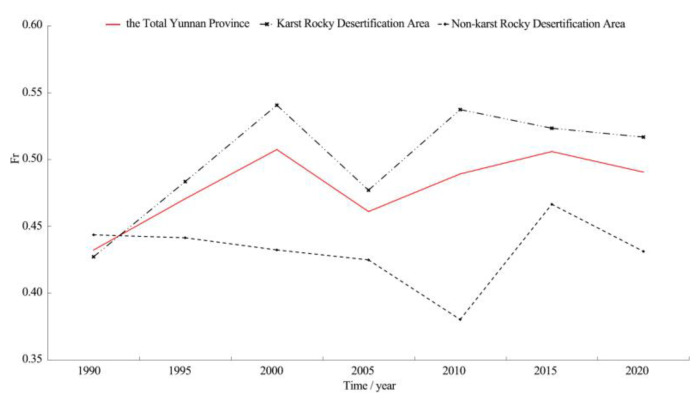
Statistical changes in the mean of Fr in Yunnan Province (1990–2020).

**Figure 7 ijerph-19-16904-f007:**
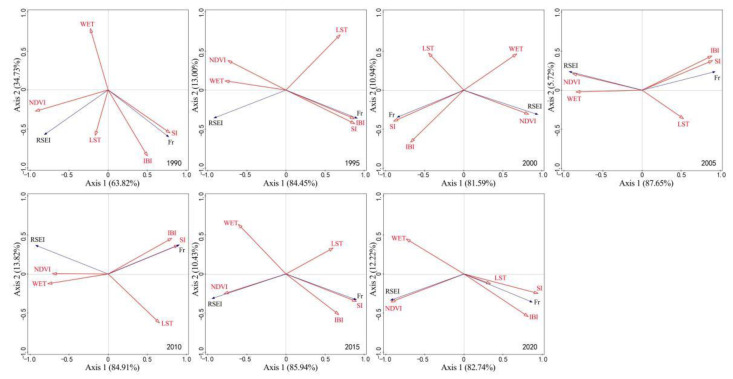
Correlation of RSEI and Fr with modeling factors. Note: RSEI, Fr, NDVI, WET, IBI, SI, and LST respectively represent remote sensing-based ecological index, rock outcrop rate, normalized vegetation index, wetness component, index-based built-up index, bare soil index, and land surface temperature; RSEI and Fr are response variables and the rest are interpretation variables.

**Figure 8 ijerph-19-16904-f008:**
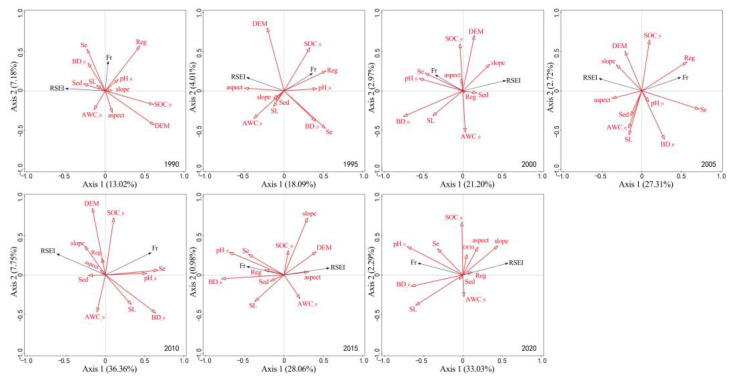
Correlation of RSEI and Fr with soil and topographic factors. Note: RSEI, Fr, dem, slope, aspect, pH__0_, BD__0,_ AWC__0_, SOC__0_, Se, Reg, SL, and Sed respectively represent remote sensing-based ecological index, rock outcrop rate, elevation, slope gradient, slope aspect, soil pH value, soil bulk density, soil available water content, soil organic carbon in soil surface, soil erosion, regolith thickness, soil layer thickness, and sedimentary layer thickness; RSEI and Fr are response variables and the rest are interpretation variables.

**Figure 9 ijerph-19-16904-f009:**
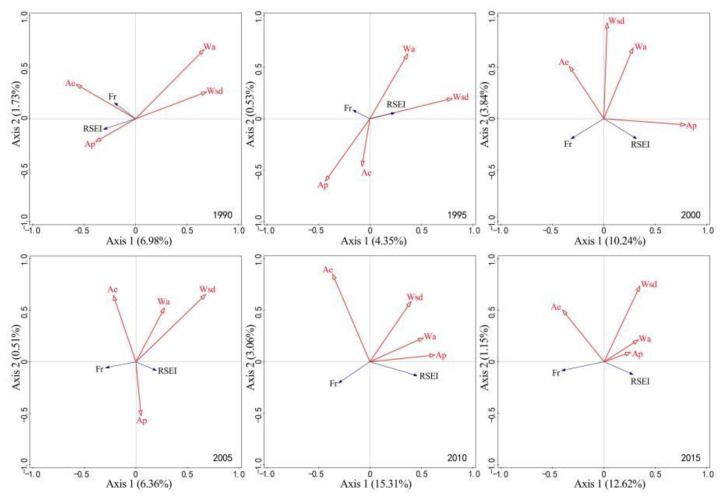
Correlation of RSEI and Fr with water factors. Note: RSEI, Fr, Ae, Ap, Wa, and Wsd respectively represent remote sensing-based ecological index, rock outcrop rate, annual actual evapotranspiration, annual precipitation, water body area, and water surface density; RSEI and Fr are response variables and the rest are interpretation variables.

**Figure 10 ijerph-19-16904-f010:**
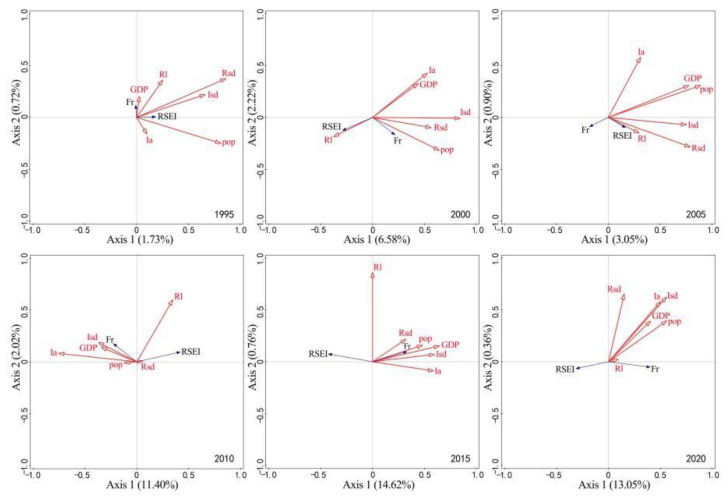
Correlation of RSEI and Fr with anthropogenic factors. Note: RSEI, Fr, pop, GDP, Rl, Rsd, Ia, and Isd respectively represent remote sensing-based ecological index, rock outcrop rate, population spatial distribution, GDP spatial distribution, road length, road surface density, impervious surface area, and impervious surface density; RSEI and Fr are response variables and the rest are interpretation variables.

**Table 1 ijerph-19-16904-t001:** Main data sources and basic information.

Data Type	Data Sources	Basic Information
Landsat images TM/OLI	Geo-spatial cloud	Spatial resolution of 30 m
Digital elevation GDEMV2	Geo-spatial cloud	Spatial resolution of 30 m, indicators include elevation, slope gradient, slope aspect
Soil texture gridded data	Hengl and MacMillan, 2019 [29]	Spatial resolution of 250 m, soil surface indicators include pH value, soil bulk density, soil available water content, soil organic carbon
A gridded global dataset of soil, intact regolith, and sedimentary deposit thicknesses	The U.S. Department of Energy’s (DOE) Oak Ridge National Laboratory (ORNL) [30]	Spatial resolution of 30 arcsec, indicators include soil, regolith, and sedimentary deposit
Spatial distribution data of soil erosion in China	Resource and Environment Science Data Center (RESDC), China Academy of Sciences (CAS)	Spatial resolution of 1 km
A dataset of China 30 m resolution actual evapotranspiration	Bintao Liu’s research team, Institute of Mountain Hazards and Environment (IMHE), CAS	Spatial resolution of 30 m, unit of mm
A dataset of spatially interpolated annual precipitation in China since 1980	RESDC, CAS	Spatial resolution of 1 km, unit of mm
Waterbody data	OpenStreetMap	Indicators include area (km^2^), surface density (km^2^/km^2^)
Spatialization approach to 1 km grid population	RESDC, CAS	Spatial resolution of 1 km, unit of persons/km^2^
Spatialization approach to 1 km grid GDP	RESDC, CAS	Spatial resolution of 1 km, unit of 10,000 RMB/km^2^
Road data	OpenStreetMap	Indicators include length (km), surface density (km^2^/km^2^)
GISD30 dataset (global 30 m impervious surface dynamic) from 1985 to 2020	Liangyun Liu’s research team, CAS [31]	Spatial resolution of 30 m, indicators include area (km^2^), surface density (km^2^/km^2^)

**Table 2 ijerph-19-16904-t002:** Hierarchical Statistics of Ecological Quality in Yunnan Karst Rocky Desertification Areas (1990–2020).

Ecological Quality Grading	1990		1995		2000		2005		2010		2015		2020	
	Area/km^2^	Percentage	Area/km^2^	Percentage	Area/km^2^	Percentage	Area/km^2^	Percentage	Area/km^2^	Percentage	Area/km^2^	Percentage	Area/km^2^	Percentage
1 (RSEI ≤ 0.2)	73.95	0.03%	2.59	0.00%	59.01	0.02%	35.34	0.01%	2884.27	1.08%	366.73	0.14%	1899.21	0.71%
2 (0.4 ≥ RSEI > 0.2)	376.12	0.14%	58.22	0.02%	10,952.22	4.11%	149.64	0.06%	102,920.89	38.60%	1432.21	0.54%	1869.48	0.70%
3 (0.6 ≥ RSEI > 0.4)	11,834.39	4.44%	89,928.48	33.74%	171,453.79	64.31%	52,405.77	19.66%	137,727.99	51.66%	110,170.20	41.32%	30,611.90	11.48%
4 (0.8 ≥ RSEI > 0.6)	95,999.08	36.01%	176,517.59	66.23%	83,748.57	31.41%	213,379.71	80.03%	22,271.61	8.35%	148,742.25	55.79%	114,324.74	42.88%
5 (RSEI > 0.8)	158,336.15	59.39%	33.16	0.01%	405.87	0.15%	649.26	0.24%	816.15	0.31%	5909.20	2.22%	117,915.04	44.23%

**Table 3 ijerph-19-16904-t003:** Hierarchical statistics of rocky desertification degree in Yunnan karst rocky desertification areas (1990–2020).

Rocky Desertification Degree	1990		1995		2000		2005		2010		2015		2020	
	Area/km^2^	Percentage	Area/km^2^	Percentage	Area/km^2^	Percentage	Area/km^2^	Percentage	Area/km^2^	Percentage	Area/km^2^	Percentage	Area/km^2^	Percentage
1(Fr ≤ 20%)	61,862.95	23.20%	53,900.32	20.22%	49,976.78	18.74%	62,861.94	23.58%	51,503.26	19.32%	49,346.78	18.51%	42,318.53	15.87%
2(20% < Fr ≤ 30%)	38,650.57	14.50%	29,775.82	11.17%	20,812.68	7.81%	28,436.86	10.67%	23,380.62	8.77%	25,367.50	9.51%	33,609.45	12.61%
3(30% < Fr ≤ 50%)	74,597.15	27.98%	59,438.56	22.30%	46,427.35	17.41%	52,561.97	19.71%	45,428.02	17.04%	50,992.26	19.13%	58,120.15	21.80%
4(50% < Fr ≤ 70%)	40,413.82	15.16%	52,417.93	19.67%	52,560.88	19.71%	49,122.37	18.42%	49,095.68	18.41%	52,774.28	19.79%	47,791.75	17.93%
5(70% < Fr ≤ 90%)	26,646.83	9.99%	41,073.59	15.41%	55,073.82	20.66%	42,599.57	15.98%	54,267.71	20.35%	51,908.41	19.47%	49,498.93	18.57%
6(Fr > 90%)	24,447.10	9.17%	29,933.20	11.23%	41,767.17	15.67%	31,036.02	11.64%	42,945.04	16.11%	36,229.18	13.59%	35,247.27	13.22%

## Data Availability

Not applicable.

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
