# Peer review of "Ecological Quality Evolution and Its Driving Factors in Yunnan Karst Rocky Desertification Areas"

_ijerph, 2022, doi:10.3390/ijerph192416904_

Round 1

Reviewer 1 Report

Dear Authors,

Your work is very interesting, but You muast add few information to Introduction and Conclusion. A few figures are small resolution, please change this resolution. My sugestion are in pdf file.

Introduction - add references.

Methods and materials - very good. 

Results - very good.

Discusion - I think, You must add information from other places on the world. Are You first reserchers wich write about this problem?

Conclusion - add information.

Best regards

Reviewer

Reviewer 2 Report

Introduction

A brief approach should be made to the factors used in the analysis and their role in desertification

Material & Methods

Improve the visibility of Figure 1 (increase the font size)

Make a brief characterization of the region under study (soils, climate, ...).

It would be useful to validate the models used! Is it possible to do that?

Which Key driving factors were used?

Results

It is necessary to greatly improve the visibility, quality, and information available in Figure 3 to Figure 10. In some Figures it is not possible to read the legends!

Point 3.2.1 - you should delete the first two sentences ("Rocky desertification is the extreme form of land degradation [25-26]. It is represent by serious soil erosion, outcrops in large areas and a dramatic decline in land productivity under the condition of surface vegetation damage [27]"). This section is for the presentation of the results!

Figures 3 and 5 should include an indication of the geographical areas analyzed (Southeast Yunnan, ...)

Point 3.2.1 - It would be interesting to specify the characteristics of the degree of rocky desertification

The use of acronyms, without the proper meaning, in the legends of Figures and Tables, should be avoided

Some acronyms do not have the meaning

pH must be written with lowercase p. You must correct throughout the text

Round 2

Reviewer 2 Report

I have no additional comments.